# Understanding spiking networks through convex optimization

**Allan Mancoo** *
Champalimaud Centre for the Unknown, Lisbon, Portugal
Ecole Normale Superieure, Paris, France
allan.mancoo@neuro.fchampalimaud.org

**Sander W. Keemink** *
Champalimaud Centre for the Unknown, Lisbon, Portugal
sander.keemink@research.fchampalimaud.org

**Christian K. Machens**
Champalimaud Centre for the Unknown, Lisbon, Portugal
christian.machens@neuro.fchampalimaud.org

## Abstract

Neurons mainly communicate through spikes, and much effort has been spent to understand how the dynamics of spiking neural networks (SNNs) relates to their connectivity. Meanwhile, most major advances in machine learning have been made with simpler, rate-based networks, with SNNs only recently showing competitive results, largely thanks to transferring insights from rate to spiking networks. However, it is still an open question exactly which computations SNNs perform. Recently, the time-averaged firing rates of several SNNs were shown to yield the solutions to convex optimization problems. Here we turn these findings around and show that virtually all inhibition-dominated SNNs can be understood through the lens of convex optimization, with network connectivity, timescales, and firing thresholds being intricately linked to the parameters of underlying convex optimization problems. This approach yields new, geometric insights into the computations performed by spiking networks. In particular, we establish a class of SNNs whose instantaneous output provides a solution to linear or quadratic programming problems, and we thereby reveal their input-output mapping. Using these insights, we derive local, supervised learning rules that can approximate given convex input-output functions, and we show that the resulting networks are consistent with many features from biological networks, such as low firing rates, irregular firing, E/I balance, and robustness to perturbations and synaptic delays.

## 1 Introduction

The brain functions through vast networks of neurons, which mainly interact through sparse and irregular spiking activity. Yet even today, much of both neuroscience theory (e.g. [1, 2, 3]) and machine learning [4] relies on rate-based units. Learning and understanding the core computations of SNNs have proven more challenging. Only recently have SNNs shown competitive results in machine learning, partly through new, spike-based learning rules [5, 6, 7], but mainly through transferring insights from rate to spiking networks [8, 9, 10].

---

A key hurdle has been that spiking networks are hard to treat analytically. Major insights have often been limited to bottom-up approaches based on randomly connected networks [11, 12, 13], or to single neuron computations based on spike-timing [14, 15, 16]. An alternative way to understand SNN computations has been to derive spiking behavior directly from some loss functions with biology-inspired constraints [17, 18, 19, 20]. Using this approach, it has been shown that the time-averaged activity of some SNNs can solve some underlying convex optimization problem such as quadratic programs (QPs) [21, 22] or linear programs (LPs) [23].

Here, we build on these approaches, as well as some recent geometric insights [24], and show that virtually all input-driven (or inhibition-dominated) SNNs are intimately tied to convex optimization problems. We link the connectivity, thresholds, and time-scales of SNNs to the parameters of LPs and QPs.

Using these insights, we next clarify the input-output functions of all SNNs that solve convex optimization problems. We show geometrically that such SNN layers effectively compute convex, piecewise-linear functions, and thus can approximate any convex input-output function. The computational power of 'convex layers' has previously been demonstrated in a pure machine-learning context [25, 26], and without a specific network implementation in mind. We here show that SNNs provide a natural implementation for such layers. We additionally derive local learning rules to implement a given computation. Given that the resulting spiking network layers show many features from biological networks (such as irregular spiking, E/I-balance, and robustness to perturbations and synaptic delays), we propose that such layers are, in fact, closer to biology than the standard ReLu layers.

## 2 Spiking neural networks and convex optimization

### 2.1 Spiking neural network models

Spiking neural networks (SNNs) can be modeled with a broad range of neuron models. Here, we focus on arguably the most common model, leaky integrate-and-fire (LIF) neurons. A network of $N$ such neurons is described by their voltage dynamics,

$$\dot{\mathbf{V}}(t) = -\lambda \mathbf{V}(t) + \mathbf{F}\mathbf{c}(t) + \mathbf{\Omega}\mathbf{s}(t) + \mathbf{I}_{bg}(t), \tag{1}$$

where $\mathbf{V}(t) \in \mathbb{R}^N$ are the neurons' voltages, $\lambda$ determines the membrane leak time-constant, $\mathbf{c}(t) \in \mathbb{R}^K$ are the inputs, $\mathbf{F} \in \mathbb{R}^{N \times K}$ are the forward weights, $\mathbf{\Omega} \in \mathbb{R}^{N \times N}$ are the recurrent weights, $\mathbf{s}(t) \in \mathbb{R}^N$ are the neural spike trains, and $\mathbf{I}_{bg}(t) \in \mathbb{R}^N$ are background currents or noise. A spike is generated whenever a neuron's voltage crosses its threshold, $T_i$, and a spike train is described as a sum of delta-functions, $s_i(t) = \sum_{t_j} \delta(t - t_j)$. Whenever a neuron spikes, its voltage is reset to a resting potential, $R_i$, which here is implicitly implemented in the all-negative diagonal terms of $\mathbf{\Omega}$, so that $R_i = T_i - \Omega_{ii}$.

### 2.2 Linear and quadratic programming and their geometry

Our key insight here is that one can examine the dynamics and computations of SNNs through the perspective of a convex optimization problem with inequality constraints. Inequality constraints have a natural correlate in spiking networks, as the voltages of neurons are always bound to be less than their thresholds. We start with a fairly generic optimization problem,

$$\begin{aligned} \underset{\mathbf{y}}{\text{Minimize}} \quad & \left( E(\mathbf{y}) = \frac{\lambda}{2}\mathbf{y}^\top\mathbf{y} + \mathbf{b}^\top\mathbf{y} \right) \\ \text{subject to} \quad & \mathbf{F}\mathbf{x} - \mathbf{G}\mathbf{y} \leq \mathbf{T} \end{aligned} \tag{2}$$

where $\mathbf{y} \in \mathbb{R}^M$ is the optimization variable, $\mathbf{b}$ is a bias, and $\mathbf{x} \in \mathbb{R}^K$ is an input to the problem. The remaining variables, $\mathbf{F} \in \mathbb{R}^{N \times K}, \mathbf{G} \in \mathbb{R}^{N \times M}, \mathbf{T} \in \mathbb{R}^N$ and the positive scalar $\lambda$ are parameters of the optimization problem. This type of optimization problem is generally referred to as a quadratic program (QP) or, if $\lambda = 0$, a linear program (LP) [27].

Before mapping this optimization problem onto the spiking network, equation (1), we will review its geometric interpretation. We can rewrite the $i$-th inequality constraint as $\mathbf{G}_i^\top \mathbf{y} \geq \mathbf{F}_i^\top \mathbf{x} - T_i$, where the (column) vectors $\mathbf{F}_i$ and $\mathbf{G}_i$ refer to the $i$-th rows of $\mathbf{F}$ and $\mathbf{G}$, respectively. Each inequality

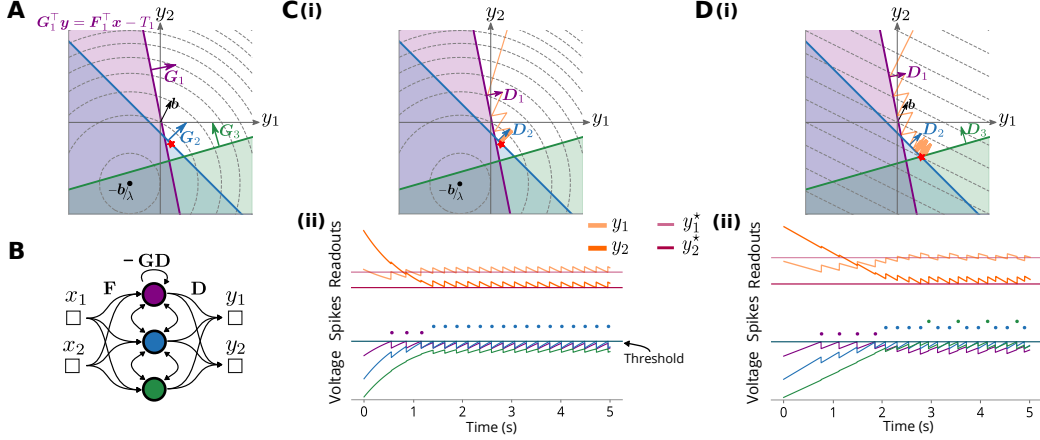

Figure 1: Convex optimization and SNNs. (A) Illustration of a quadratic program. Dashed grey circles are the contours of the objective. Colored region is the infeasible set. (B) Illustration of a network with 3 neurons. (C) Spiking network solving a quadratic program. (i) The trajectory of the network's readout (orange) bounces back into the feasible set each time it hits a boundary. (ii) Neural activity over time with corresponding readouts jumping about the QP solution, spikes and voltages. (D) Same as (C) but the spiking network is now solving a linear program.

constraint thereby divides the $\mathbf{y}$-space (where the optimization variable resides) into two half-spaces, with boundary defined at equality. Fig. 1A shows an example for a two-dimensional optimization variable ($M = 2$) with three constraints ($N = 3$). Each colored region shows the half-space where the corresponding inequality constraint is violated. The union of colored half-spaces thus defines the infeasible set for the optimization problem, and the white area the feasible set. The QP solution, $\mathbf{y}^\star$, corresponds to the minimum of the objective function within the feasible set (Fig. 1A, red star).

## 2.3 Gradient descent under constraints

Several highly efficient algorithms exist to solve QPs [27, 28]. Here, we start with a simple algorithm: gradient descent. In continuous time, minimizing the objective function, equation (2), yields the differential equation

$$\dot{\mathbf{y}} = -\frac{\partial E}{\partial \mathbf{y}} = -\lambda \mathbf{y} - \mathbf{b}. \tag{3}$$

To avoid that $\mathbf{y}$ leaves the feasible set, we will assume that each boundary reflects it into a direction $\mathbf{D}_i$. By modeling each such bounce as a delta-function, we obtain the optimization dynamics

$$\dot{\mathbf{y}} = -\lambda \mathbf{y} - \mathbf{b} + \mathbf{D}\mathbf{s}(t), \tag{4}$$

where $\mathbf{s}(t)$ is the $N$-dimensional vector of bouncing events, similar to the spike trains introduced above (equation 1), and $\mathbf{D} \in \mathbb{R}^{M \times N}$ is the matrix of bounce directions, with columns $\mathbf{D}_i$ [2].

The direction of each bounce is, of course, crucial to the final solution. For now, we will simply assume that $\mathbf{y}$ always bounces back orthogonal to the respective boundary, so that $\mathbf{D}_i \propto \mathbf{G}_i$. This condition is sufficient (but not necessary) to guarantee that $\mathbf{y}$ approaches the minimum of the loss function within the feasible set. Eventually, $\mathbf{y}$ approximates the true solution to the QP with a discretization error, $\eta$, which depends on the size of the jumps $\mathbf{D}_i$. We illustrate this in Fig. 1(C, D), where the orange trajectory shows the dynamics of $\mathbf{y}$ due to descent on the loss function, interspersed with bouncing events, whenever one of the boundaries is hit. Once close to the solution (red star), the dynamics jumps back and forth around the true solution. If we set the parameter $\lambda = 0$, the dynamics moves $\mathbf{y}$ close to the solution of a linear program (LP), see Fig. 1D [27]. In contrast to Fig. 1C, the dynamics is now driven by a constant drift, $-\mathbf{b}$, rather than exponential decay.

## 2.4 From convex optimization to the voltage dynamics of LIF neurons

To link this optimization problem to SNNs, we first define the left-hand-side of the inequality constraint, equation (2), as the voltages of the neurons, $\mathbf{V} = \mathbf{Fx} - \mathbf{Gy}$, and the parameters $\mathbf{T}$ on the right-hand-side as their thresholds. Furthermore, we assume that the input $\mathbf{x}$, just as $\mathbf{y}$, is a time-dependent quantity. Taking the temporal derivative of this voltage, we obtain

$$\dot{\mathbf{V}} = \mathbf{F}\dot{\mathbf{x}} - \mathbf{G}\dot{\mathbf{y}} \tag{5}$$

$$= \mathbf{F}\dot{\mathbf{x}} + \lambda\mathbf{Gy} + \mathbf{Gb} - \mathbf{GDs}(t) \tag{6}$$

$$= -\lambda\mathbf{V} + \mathbf{F}(\lambda\mathbf{x} + \dot{\mathbf{x}}) - \mathbf{GDs}(t) + \mathbf{Gb}. \tag{7}$$

Equation (6) follows from equation (5) by replacing $\dot{\mathbf{y}}$ by its dynamics (4), and equation (7) follows from (6) by using the definition of the voltage.

We note that the resulting voltage dynamics now corresponds to a network of LIF neurons, equation (1). The parameter $\lambda$, tied to the quadratic loss above, determines the membrane leak time-constant, the matrix $\mathbf{F}$ has become the forward weight matrix with inputs $\mathbf{c}(t) = \lambda\mathbf{x} + \dot{\mathbf{x}}$, the matrix $\mathbf{G}$, together with the matrix of bouncing directions, $\mathbf{D}$, has become the recurrent weight matrix, $\boldsymbol{\Omega} = -\mathbf{GD}$ (with rank given by $M$, the dimensionality of $\mathbf{y}$). The bouncing events, $\mathbf{s}(t)$, are now the spike trains, and the bias $\mathbf{b}$ is an external current fed to the network via the weights $\mathbf{G}$. In other words, the resulting SNN directly implements the gradient optimization described above, and solves quadratic programs (if voltages leak, i.e., $\lambda > 0$), or linear programs (in the absence of a voltage leak, i.e., $\lambda = 0$), up to a discretization error, $\eta$.

If we define the instantaneous firing rates of neurons, $\mathbf{r}(t)$, as filtered versions of their spike trains, i.e., $\dot{\mathbf{r}} = -\lambda\mathbf{r} + \mathbf{s}(t)$, then our optimization variable satisfies $\mathbf{y} = \mathbf{Dr} - \frac{\mathbf{b}}{\lambda}$, which can be interpreted as the instantaneous readout by a downstream layer. We illustrate the behavior of the network in Fig. 1(C, D) for a QP and LP problem, respectively.

However, we can also free ourselves from minimizing the optimization problem, and consider what happens if we choose the bouncing directions $\mathbf{D}$ differently. First, we could simply choose $\mathbf{D}$ to keep $\mathbf{y}$ within the feasible set. A necessary (but not sufficient) condition for the jump to return to the feasible set is that $\Omega_{ii} = -\mathbf{G}_i^\top\mathbf{D}_i \leq 0$, which simply corresponds to the requirement that a neuron's self reset (which we model through the diagonal entries of $\boldsymbol{\Omega}$) is negative. A sufficient (but not necessary) condition for each jump to return to the feasible set is that $\Omega_{ij} = -\mathbf{G}_i^\top\mathbf{D}_j \leq 0$, which corresponds to the constraint that all recurrent connections are inhibitory. In all cases, the dynamics of the network will effectively wander along the boundary of the feasible set, and this boundary can therefore be thought of as the manifold on which the SNN dynamics evolves.

Excitatory recurrent connections can be chosen to move the solution outside of the feasible set. Such excitations can either decay quickly (if the next neuron moves the solution back into the feasible set), or they may self-sustain (if neurons keep pushing the solution out of the feasible set). We will not cover such self-sustained solutions in this paper, and rather focus on 'input-driven' networks that solve convex optimization problems, which we define as networks that keep the variable $\mathbf{y}$ within the feasible set and close to the optimum.

## 2.5 Example networks

Several previously proposed SNNs fit into this framework. Here, we list a few (in all cases setting $\mathbf{b} = 0$):

**ReLu Layer.** If we assume independent neurons, we get $M = N$, and $\mathbf{G} = \mathbf{D}^\top = \mathbf{I}$. In this case, $\mathbf{y} = \mathbf{r}$ and the voltages become $\mathbf{V} = \mathbf{Fx} - \mathbf{r} \leq \mathbf{T}$, which we can re-write as $\mathbf{r} \geq \mathbf{Fx} - \mathbf{T}$. Since the objective of the optimization problem is quadratic, we find that the SNN dynamics leads to the solution

$$\mathbf{r} = \max(\mathbf{Fx} - \mathbf{T}, \mathbf{0}) + \eta. \tag{8}$$

**Spike Coding.** If we assume $\mathbf{G} = \mathbf{F}$, the voltages then become $\mathbf{V} = \mathbf{Fx} - \mathbf{FDr} \leq \mathbf{T}$ which is the voltage definition of the 'spike coding' networks learnt in [29]. These networks have been shown to generate biologically realistic activity (asynchronous, irregular spiking, balance of excitation and inhibition).

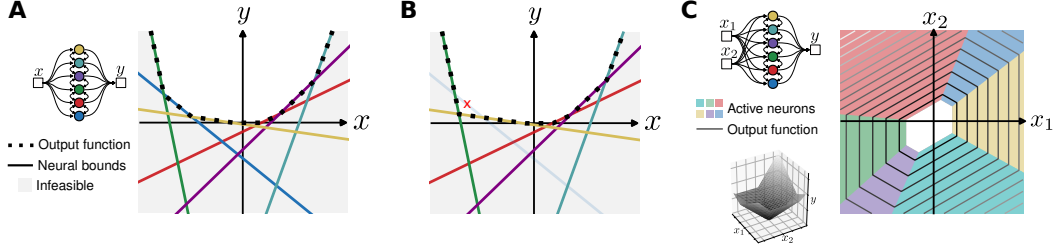

Figure 2: Interpreting spiking network computations. (A) Example network of six interconnected neurons performing a one-dimensional transformation. The corresponding input-output function is illustrated on the right in $(x, y)$-space. Solid lines illustrate neural boundaries, and the dotted line the resulting input-output functions (minus the discretization error $\eta$). (B) The effect of changing or silencing a single neuron (red cross) on the input-output function. (C) Same as in (A), but for a 2D input. Now each neuron's inequality boundary corresponds to a plane in 3D space. Different neurons are active in different parts of this space (colored patches), and locally determine the slope of the output function (as shown by the contours). White corresponds to having no neurons active. Inset shows the output function in 3D.

**Sparse Coding.**  If we additionally assume $\mathbf{G} = \mathbf{F} = \mathbf{D}^\top$, the voltages become $\mathbf{V} = \mathbf{D}^\top \mathbf{x} - \mathbf{D}^\top \mathbf{D} \mathbf{r} \leq \mathbf{T}$, and we recover previously proposed auto-encoder SNNs [21, 18]. Interestingly, with these parameter choices, the 'argmin' solution to the dual of the quadratic programming problem (using Lagrangian parameters $\mathbf{r} \geq 0$), is given by [3]

$$\mathbf{r}^\star = \arg\min_{\mathbf{r} \geq \mathbf{0}} ||\mathbf{x} - \mathbf{D}\mathbf{r}||^2 + \mathbf{r}^\top \mathbf{T}, \tag{9}$$

which is the classical sparse coding loss function [30]. We note that the thresholds now explicitly appear in the objective and effectively implement a sparsity cost on the firing rates. Although this feature was implicitly captured in previous SNNs, we here provide a more formal understanding of the sparsity effects of the spiking thresholds.

## 3 Understanding computations in SNN layers

Given the link between SNNs and convex optimization, we now consider the possible computations done by such networks. More specifically, we will study how the output $\mathbf{y}$ depends on the input $\mathbf{x}$. For simplicity, we will assume $\mathbf{b} = 0$. To guarantee that the SNNs approximate the optimum, we choose $\mathbf{D}_i \propto \mathbf{G}_i$ and $G_{ij} \geq 0$ (see Section 2.4). This choice of $\mathbf{G}$ ensures that for all inputs $\mathbf{x}$, there is a feasible set, thus avoiding ill-defined cases.

### 3.1 One-dimensional outputs

If we set $M = 1$, each neuron's voltage inequality becomes $\mathbf{F}_i^\top \mathbf{x} - G_i y \leq T_i$. Solving for $y$, we obtain the set of inequalities $y \geq (\mathbf{F}_i^\top \mathbf{x} - T_i)/G_i$. Since the dynamics of the spiking network minimizes the loss $L = \lambda y^2/2$, the final solution will be either $y = 0$, or be bouncing off the spiking boundary of active neuron $j$, given by $y = (\mathbf{F}_j^\top \mathbf{x} - T_j)/G_j$. We can therefore write the input-output function as

$$y = f(\mathbf{x}) = \texttt{max}\Big(0, \frac{\mathbf{F}_1^\top \mathbf{x} - T_1}{G_1}, \frac{\mathbf{F}_2^\top \mathbf{x} - T_2}{G_2}, ...\Big) + \eta, \tag{10}$$

which is effectively a piecewise-linear function which partitions the space according to a set of linear equations (see Fig. 2A). Each neuron's boundary determines the output function locally, until either $y = 0$ or another neuron becomes active instead. Importantly, each neuron only acts within a limited

local range, which endows the network with a certain inherent robustness to single neuron changes in parameters and even cell death, as this would only locally affect the input-output function (Fig. 2B). Such computational units, called 'maxout' units, have previously been suggested as advantageous in the machine-learning literature and can be universal approximators [31].

## 3.2 M-dimensional outputs

Conceptually, the extension to $M$-dimensional outputs is straightforward. A given neuron's spiking boundary is described by all points satisfying $\mathbf{F}_i^\top \mathbf{x} - \mathbf{G}_i^\top \mathbf{y} = T_i$, which can be viewed as a $(K + M - 1)$-dimensional hyperplane in the $(K + M)$-dimensional space defined by $\mathbf{u} = (\mathbf{x}, \mathbf{y})$. As before, each neuron's hyper-plane divides the space into a feasible and infeasible region. The solution is constrained to lie either at $\mathbf{y} = 0$ or be bouncing off one (or more) of these hyperplanes. Consequently, the solution is a piecewise-linear, convex function of the input $\mathbf{x}$.

# 4 A geometric view on supervised learning in SNNs

The above considerations have given us a better understanding of the computations performed by a single layer of spiking neurons with recurrent inhibitory connections. We will now study how these insights reshape our view of supervised learning problems.

## 4.1 Learning through basis functions

The classical neural network learning approach is to consider the output of the neurons at a given layer to be a set of basis functions, which can then be combined linearly for arbitrary input-output transformations by training the output weights [4, 8]. The same applies within our framework. By using random parameters $\mathbf{F}$, $\mathbf{G}$, $\mathbf{D}$, and $\mathbf{T}$ (properly constrained to produce feasible sets) a rich set of $M$ convex basis functions can be generated as the $M$-dimensional output of a network. If the set is rich enough, these basis functions can then be combined to approximate arbitrary, non-convex input-output functions. Geometrically, this approach corresponds to having a fixed set of $N$ neural hyper-planes in an $M$-dimensional space (Fig. 2C), and requires a high dimensionality (in both the number of neurons and the read-out space) to generate a rich enough basis-set. However, instead of following this standard route, we will here investigate the problem of directly adjusting the neural hyper-planes themselves in order to learn specific (convex) input-output functions.

## 4.2 Learning the neural hyper-planes

We will first consider how to adjust the neural hyperplanes (that is, the corresponding inequalities) to make the network's output, $\mathbf{y}$, match a desired target, $\tilde{\mathbf{y}}$. Previous approaches used gradient descent on a loss function defined over a global error [25, 26, 32], leading to highly non-local parameter updates, which is unlikely to be used in biological networks (similar arguments hold for iterative repartitioning algorithms [33, 34]). From the geometric interpretation of the problem, we note three facts. First, individual neurons do not have direct access to the global variable, $\mathbf{y}$, or to the error, $\mathbf{e} = \mathbf{y} - \tilde{\mathbf{y}}$, but rather get indirect information through a projection on their encoder weights, $\mathbf{G}_i$. Second, at spike-time this projection satisfies $\mathbf{G}_i^\top \mathbf{y} = \mathbf{F}_i^\top \mathbf{x} - T_i$ which is the boundary equation (see Fig. 1). Third, the end result of learning should be that each boundary locally supports the epigraph of some target function in $(\mathbf{x}, \mathbf{y})$-space, and that the output function is properly distributed across neurons (e.g. see Fig. 2A).

From these insights, we propose the following learning scheme. We allow neurons to drift their boundaries slowly to the epigraph by lowering their thresholds, $\dot{T}_j = -\lambda_T, \quad \forall j \in 1, \ldots, N$, where $\lambda_T$ determines the speed of the drift. Neurons, whose boundaries are far away from the targets, thereby eventually approach the epigraph of the target function from below, start spiking, and thus take part in the learning. At the same time, we let all spiking neurons adjust their boundaries and redistribute them along the epigraph. To do so, we take a simple approach and minimize the projected error on each neuron's encoding weights. For one data point, we therefore minimize the loss

$$L = \frac{1}{2} \sum_i^{N_{act}} ||\mathbf{G}_i^\top (\mathbf{y} - \tilde{\mathbf{y}})||^2 = \frac{1}{2} \sum_i^{N_{act}} ||\mathbf{F}_i^\top \mathbf{x} - T_i - \mathbf{G}_i^\top \tilde{\mathbf{y}}||^2 \tag{11}$$

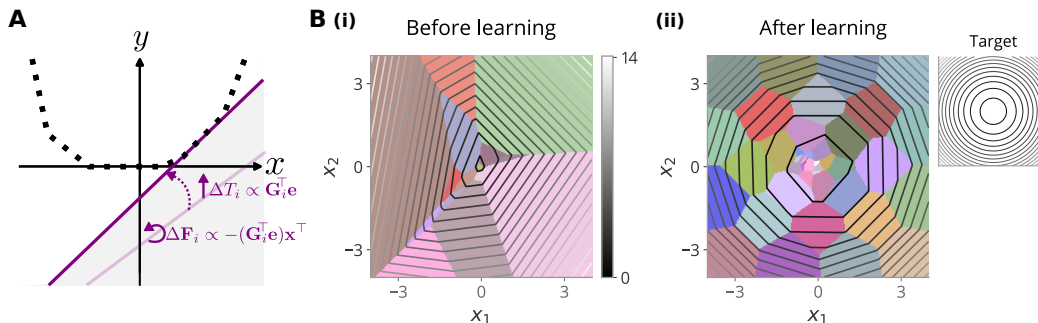

Figure 3: Supervised learning of spiking network. (A) Illustrating the effect of learning the feed-forward weights, $\mathbf{F}$ and thresholds $\mathbf{T}$ in (x, y)-space. Learning $T_i$ for neuron $i$ translates its spiking boundary as an offset and learning $\mathbf{F}_i$ changes its slope. With competition across neurons, the violet neuron only learns to fit data points in its partition. (B) Learning a paraboloid for N=50 neurons. (i) The contour plot of network output before learning with background color coded according to which neurons are active. If no neuron participates, the partition is colored white. (ii) Contour plot after learning. It is a piece-wise linear approximation of the target function shown in the inset. Source code is available at https://github.com/machenslab/spikes.

where $N_{act}$ is the subset of neurons that are actively constrained and emit spikes for a given input-output pair. Here, we focus only on the forward weights, $\mathbf{F}$, and thresholds, $\mathbf{T}$, which will be sufficient to guarantee some distribution along the epigraph (more general and efficient learning rules would include changes in the recurrent connectivity as well). By computing the corresponding gradients, we get the following rules for $i \in N_{act}$:

$$\Delta T_i = -\alpha \frac{\partial L}{\partial T_i} = \alpha \left( \mathbf{F}_i^\top \mathbf{x} - T_i - \mathbf{G}_i^\top \tilde{\mathbf{y}} \right) = \alpha \mathbf{G}_i^\top \mathbf{e} \tag{12}$$

$$\Delta \mathbf{F}_i = -\alpha \frac{\partial L}{\partial \mathbf{F}_i} = -\alpha \left( \mathbf{F}_i^\top \mathbf{x} - T_i - \mathbf{G}_i^\top \tilde{\mathbf{y}} \right) \mathbf{x}^\top = -\alpha (\mathbf{G}_i^\top \mathbf{e}) \mathbf{x}^\top, \tag{13}$$

where $\alpha$ is the learning-rate. To ensure that the drift in thresholds does not dominate the learning, we set $\lambda_T \ll \alpha$. Additionally, we introduce a cost $\mu$ on spikes to allow a proper distribution of the code across neurons. This limits arbitrarily high activity of individual neurons by lowering the reset after a spike to $R_i = T_i - \Omega_{ii} - \mu$. This cost can be thought of as a type of regularization similar to what has previously been used in SNNs [18].

We illustrate the effects of these rules in Fig. 3A. Whenever a neuron spikes, it shifts ($\Delta T_i$) and rotates ($\Delta \mathbf{F}_i$) its spiking boundary in $(x, y)$-space, thereby reaching a partition of the input-target $(x, \tilde{y})$ samples. Through many presentations of input-target pairs (and assuming that different neurons start out with different random parameters), neurons can thus jointly find a piece-wise linear fit of a target function (Fig. 2A).

### 4.3 Simple paraboloid example

To demonstrate the effect of the learning rules in a simple toy example, we trained an SNN with $N = 50$ neurons to reproduce a paraboloid in 3D-space, $\tilde{y} = 0.3(x_1^2 + x_2^2)$, as shown in the inset of Fig. 3B(ii). During a learning epoch, the inputs were randomly sampled from one hundred equally spaced points in $[-4, 4]$ for each $x$-dimension. In each trial, a sampled input-target pair $(\mathbf{x}, \tilde{y})$ was fed to the SNN for four seconds of simulation time (using the forward Euler method). To reduce the effect of spikes due to transients as the input changed across trials, we only started training 1s after the onset of the trial. We ran the algorithm for 100 epochs with each epoch covering the whole input space. Finally, we turned off the teaching signal and ran the network with learnt parameters. Fig. 3B(ii) shows the contours of the network readout (averaged over the last few time bins) which is a piecewise linear fit to the paraboloid. By color coding the background of the contour plot based on which neurons spike, we see a more distributed but still distinct partitioning of the input space after learning (contrast Fig. 3B(i) and B(ii)).

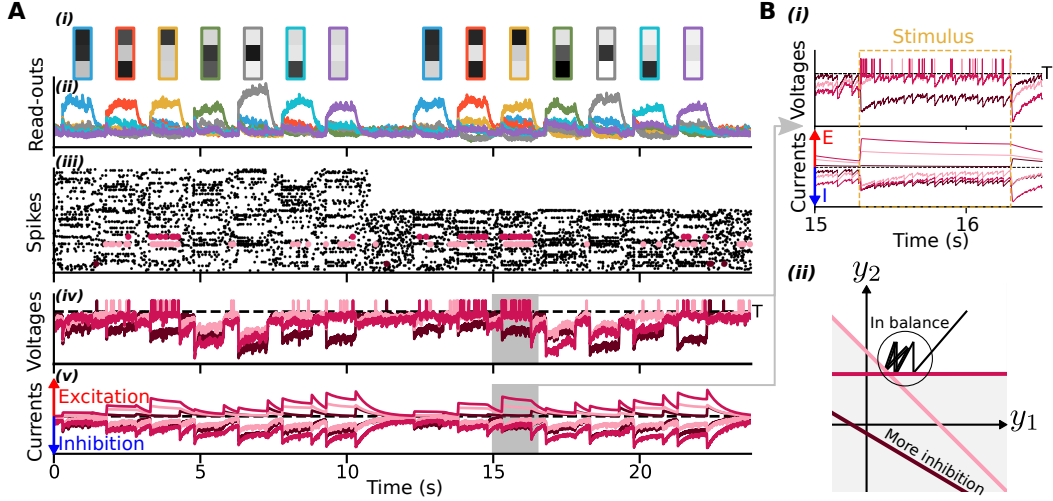

Figure 4: Spiking behavior in larger networks, illustrated with a simple classification network. (A) (i) Stimulus inputs. (ii) Network readouts, with colors corresponding to the possible input combinations. (iii) The underlying spike patterns. (iv) The voltages of the three example neurons highlighted in iii. (v) The currents affecting the example neurons. Note that for clarity these currents do not include the input or voltage noise. The gray areas correspond to panel B(i). The currents and voltages were normalized relative to each neuron's threshold. Spikes are illustrated by vertical lines. In the absence of a stimulus the mean spiking CV was $0.99 \pm 0.28$. (B) E/I balance for different neurons. (i) Voltages and currents from the gray area in panels A(iv,v). (ii) Cartoon illustrating the E/I-balance of active and inactive neurons. Black curve illustrates the readout, and the colored lines the neural boundaries. Network parameters: $N = 300$, $\lambda = 2$, $\mathbf{D} = 0.1\mathbf{G}^\top$, $\mu = 0.1$, $\sigma_V = 0.1$, synaptic delay = 2ms. Learning parameters: $\alpha = 0.1$, $\lambda_T = 0.001$, both decaying across epochs according to $\exp(-0.001 n_{\text{epoch}})$, for 750 epochs. Perturbation noise parameters: $\sigma_{\text{stim}} = 0.1$, $\sigma_{\text{OU}} = 0.05$, $\lambda_{\text{OU}} = 10$. Source code is available at https://github.com/machenslab/spikes.

## 5   A larger spiking network

Finally, we demonstrate that the resulting networks exhibit many features of biological systems when they are scaled up. Our examples so far have focused on simple toy scenarios, in which only one or a few neurons are active for each possible input, resulting in regular spiking activity (Fig. 1C). This is no longer true when we use several readout dimensions ($M > 1$). For example, for $M = 2$, we can already find many boundary intersection points, in which case two neurons are active simultaneously (as in Fig. 1D). For higher dimensions, this quickly leads to quite complex spiking behavior.

We demonstrate these observations with a network trained to classify a vector of three input pixels ($K = 3$) (Fig. 4). The readouts were 7-dimensional ($M = 7$), with a particular read-out jumping up when a corresponding pixel combination was detected. In the absence of any inputs, the read-outs are all equal. Throughout all simulations we added small amounts of Gaussian noise to the voltages. To test the robustness of the resulting network, we added several perturbations not present during learning. First, each input pixel was corrupted by a random offset drawn from a Gaussian distribution on each trial, as well as a drifting noise process over time (simulated by an Ornstein-Uhlenbeck (OU) process). Second, we added a 2ms synaptic transmission delay. Third, we silenced 40% of the neural population halfway through the simulation.

The resulting networks function in a biologically realistic regime with irregular spiking patterns, balance of excitation and inhibition, robustness to perturbations, and low firing rates (Fig. 4). The irregularity of spiking stems largely from the complex spiking dynamics around the idealized solution point, rather than the voltage noise alone. With a sufficient amount of redundancy, the network computation is robust to cell death, as eliminating neurons (and thereby boundaries) does not substantially change the feasible set (Fig. 2B, see also [35]). Whereas eliminating neurons increases

firing rates in the remaining neurons, adding neurons decreases population firing rates, and each neuron's activity can thereby be made arbitrarily sparse.

The networks also operate in a regime of balanced excitation and inhibition (Fig. 4B). In previous balanced networks performing linear computations [18], each neuron experiences similarly balanced inputs. In spiking networks performing nonlinear computation however, this is no longer the case (Fig. 4B(i)). For a given input, the readout will be bouncing between the boundaries of the neurons required to be active (Fig. 4B(ii)). As this group of neurons will be constantly near threshold, the underlying inhibition and excitation are in balance. Other neural boundaries will be away from the current readout, and thus experience comparatively more inhibition. This in line with recent work showing that balanced networks can perform nonlinear transformations if subsets of the network receive balanced inputs, whereas the rest receives more inhibition[36].

Finally, unlike previous networks [18, 35], our networks are naturally robust to delays. This is thanks to the constraint that the problem should always be well defined (i.e. $G_{ij} \geq 0$), which ensures that spurious spikes caused by delays do not cause follow-up spikes (Fig. 5).

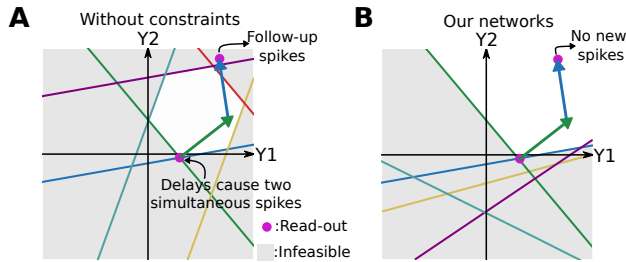

Figure 5: Illustrating the robustness to synaptic delays of our networks. (A) Without constraints on the connectivities, the feasible set described by a set of neurons quickly becomes a closed space. Delays can then cause several neurons (green + blue) to spike simultaneously, leading to multiple threshold crossings and several follow-up spikes (red + purple neurons), and so on, leading to instability. (B) If connectivity is constrained to ensure feasibility for all inputs, as in our networks, these problems are avoided. When multiple neurons fire simultaneously due to delays, the system remains within the feasible set, and hence remains stable.

## 6   Discussion and conclusions

In summary, we used a convex optimization perspective to develop a new framework to understand SNNs. We showed how a broad class of SNNs fundamentally solve QPs and LPs, expanding on what had previously been shown [21, 22, 23], but without requiring temporal integration. As a result, we gained geometric insights into the computations performed by SNNs, and we showed that they can generate piecewise-linear, convex input-output functions. Thus, we can now understand SNN layers as computing arbitrary, convex transformations of their inputs, which have been shown to be functionally more expressive than standard 'ReLu' layers [25, 26].

By focusing on input-output functions, we have largely taken a classical neural network perspective. We finally note that it may also be possible to think of SNNs and the brain as fundamentally solving convex optimization problems. Since many optimization problems can be relaxed to convex ones, this may not be a crazy thought. One intriguing observation in that context is that we reached the input-driven regime by constraining the connectivity to be all-inhibitory ($-\mathbf{G}_i^\top \mathbf{D}_j \leq 0$), which we believe is realistic from a functional point of view, as inhibition is thought to dominate during awake behavior [37]. Once the effective connectivity is inhibitory, however, the feasible set takes the shape of a cone—which relates SNNs to conic programming, a quite powerful optimization technique.

## Broader Impact

We see two possible impacts of our work. First, the use of low-powered computer chips inspired by spiking neurons (neuromorphic engineering) has been seeing a boom in technologies recently [38].

The work we present here will help these technologies as it shows how to design robust and spike-efficient networks that perform interesting computations. Second, the field of convex optimization may benefit from this work, as we show that convex optimization techniques may be directly applicable to neural systems.

Direct potential negative impacts are harder to gauge, due to the relatively local nature of this type of publication. Any negative impact will likely be found in the aggregate of many studies, rather than any specific study (exceptions, of course, exist). In such an aggregate, we see potential for danger and misuse of powerful machine-learning algorithms on low-powered devices, as our understanding of using low-powered spiking networks grows. Mass use of low-powered devices could worsen existing problems, both unintentionally (such as biased functions due to biased datasets, e.g. [39]), and intentionally (such as weaponry and surveillance applications, potentially trained by such biased datasets). These problems are worsened by the fact that many deep networks are effectively black boxes, for which we don't understand their core computations. As such, in the long term, our study might also help alleviate these types of problems as our contribution helps to understand the inner workings of SNNs.

## Acknowledgments and Disclosure of Funding

We thank Nuno Calaim for fruitful discussions and early work on the geometric interpretation. This work was supported by "Bourse d'études du Gouvernement français"/Ministry for Europe and Foreign Affairs/Embassy of France in Mauritius (AM), the Fundação para a Ciência e a Tecnologia (032077), and the Simons Collaboration on the Global Brain (543009). The authors declare no competing financial interests.

## Footnotes

[2]Here, $\mathbf{D}_i$ refers to the $i$-th column of $\mathbf{D}$ while $\mathbf{F}_i$ and $\mathbf{G}_i$ are the $i$-th rows of $\mathbf{F}$ and $\mathbf{G}$, but expressed as column vectors.

[3]With these parameter choices and $\lambda = 1$, using Lagrange multipliers $\mathbf{r} \geq 0$, the Lagrangian dual function of Eq. (2) becomes $g(\mathbf{r}) = \inf_{\mathbf{y}} \left[ \frac{1}{2} \mathbf{y}^\top \mathbf{y} + \mathbf{r}^\top (\mathbf{D}^\top (\mathbf{x} - \mathbf{y}) - \mathbf{T}) \right]$. At its minimum, we obtain $\mathbf{y} = \mathbf{D}\mathbf{r}$. Plugging this back into $g(\mathbf{r})$ yields the dual problem $\left( \min_{\mathbf{r}} \frac{1}{2} \mathbf{r}^\top \mathbf{D}^\top \mathbf{D} \mathbf{r} - \mathbf{r}^\top \mathbf{D}^\top \mathbf{x} + \mathbf{r}^\top \mathbf{T}, \text{ subject to } \mathbf{r} \geq 0 \right)$, which has the same 'argmin' solution as Eq. (9).

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
