[Reviews · NeurIPS 2020]

Review 1

Summary and Contributions: The authors compare the LIF neuron model of SNN with gradient decent of QPs and shows that the SNN layer computes a piecewise-linear, convex function of the input. Based on this insight, the authors introduce a new learning rule for SNN, in which the neurons are allowed to drift their boundaries slowly to the epigraph by lowering their thresholds and adjust their boundaries and redistribute them along the epigraph. This is achieved by minimizing the projected error on each neuron's encoding weights. The resulting network is consistent with many features from biological networks.

Strengths: The theoretical part is very clear and easy to follow. Modeling SNNs as QP/LPs is a quite interesting idea and makes sense to me. Moreover, the insights provided in this paper are also enlightening.

Weaknesses: 1) If I understand correctly, the discussions are basically on a single SNN layer under different output dimensions. Previous studies have shown that like ANNs, SNNs with multiple layers achieve better performance. Therefore, will the conclusions and insights still hold under the multi-layer scenario? 2) The experimental part is composed of numerical simulations, so it's still a little bit hard for me to imagine how can we implement the conclusions (e.g., the new learning scheme) in the SOTA SNN models for some more realistic tasks like image recognition. Can the authors give some explanations?

Correctness: Yes

Clarity: Yes

Relation to Prior Work: Yes

Reproducibility: Yes

Additional Feedback: There is no doubt that the idea of this work is novel and insightful for SNN understanding. However, the evaluation is a little rought at this stage and was not solved well in the rebuttal. I keep my score of 6. 1) Please refer to the Weakness 2) Typos such as: Page 1, line 30 "has been to to"


Review 2

Summary and Contributions: A framework describing the input-output relationship of a particular class of input-driven spiking neural networks is used to show the dynamical emergence of solutions to quadratic and linear programming problems in the network state. This convergence of network state to problem solutions depends upon a set-up of the spiking neural network in which the dynamics effectively carry out gradient descent upon the quadratic/linear programming problem with constraints enforced by spiking thresholds. The authors provide a geometric perspective on the effective dynamics of such a spiking neural network model and provide extensions to the system described. Finally, the authors derive a local supervised learning rule operating on the input matrix and spiking thresholds which allows such models to be learned (rather than constructed) such that they can produce approximations of arbitrary convex input-output functions.

Strengths: The spiking neural network models constructed are theoretically grounded and well described. The analyses made of these networks appear sound and complete. The geometric explanation of network behaviour is particularly insightful as is the formal description of the relationship between the network model described and existing work such as spike-coding networks and sparse-coding equivalents. The networks are also highly robust (proven by simulation of cell death and of increased noise). In general, the work provides a novel understanding of the dynamics and capabilities of such input-driven spiking neural network models and how the spiking threshold can be used to implement constraints on solutions and modified for learning. Such models are a potential avenue for future applications of spiking neural networks, an as yet relatively unexplored direction compared to the more traditional rate based networks. Of significant relevance to a community within NeurIPS attendees.

Weaknesses: Much of the network setup work in the first half of the paper is based upon the fairly well investigated spike-coding-esque networks. The novelty of the described networks are therefore difficult to establish -- having said this it is undeniable that the extended analysis and application to quadratic and linear programming problems is novel. Beyond this, the learning scheme proposed allows modification to neuron thresholds and input weight matrices but does not describe methods for modification of recurrent and readout weights. Given that the tasks tested in this paper have a low dimensional input and output signal, how this learning scheme would scale to higher-dimensional problems is unclear. Finally, previous work in such networks (such as the spike-coding literature) points out that these networks are highly brittle when delays are introduced to synaptic connections, it would be sensible to at minimum mention this limitation in the text.

Correctness: The claims and methods are correct and well presented.

Clarity: This paper is very well written with a clear and elegant explanation of the motivating theory and geometric perspective on the spiking neural networks described.

Relation to Prior Work: Well documented code is provided in the supplementary material with clear instructions for reproduction.

Reproducibility: Yes

Additional Feedback: Reference [25] has some kind of a character issue: “d\textquotesingle”


Review 3

Summary and Contributions: This work proposes a novel notion that a spiking network is performing a constrained convex optimization in particular the firing threshold could be treated as a constraint in optimization. As far as I know, the idea is novel. Moreover, the current work provides a clear geometric interpretation of the network dynamics in state space.

Strengths: The idea in this work is novel especially it interprets the spiking dynamics as a constraint in optimization.

Weaknesses: I think the analysis provided in this paper is sufficient for a NeurIPS paper.

Correctness: I have gone through all the math in the main text and believe the math derivation is correct.

Clarity: This paper is well written and structure-wise. And some key assumptions and derivations are clearly explained.

Relation to Prior Work: The authors mainly compare the current study with Ref. [21-23]. I think the author could compare other work in a broader scope. For example, the constraint in Eq. 2 is novel in helping us understand the coding mechanism in a spiking network model, while the constraint optimization is not included in two mainstream coding frameworks such as probabilistic population code and sampling-based codes. In the long run, I am very curious to know how the convex optimization, especially the constraint, in this work could be compared with the two aforementioned coding frameworks.

Reproducibility: Yes

Additional Feedback: The author only gives the condition the “bounce direction” D should satisfy (lines 119-121). I am wondering whether there is a way to determine a particular D, otherwise there will be multiple network realizations which could optimize the same objective function. Nothing to update after the rebuttal, because the author's response didn't reply to me.


Review 4

Summary and Contributions: This paper demonstrates that a model spiking neural network can be considered to solve quadratic convex optimization problems with linear constraints. The highly nonlinear (spiking) nature of these networks make them hard to analyze. Therefore, the problem is potentially of interest to the neuroscience community.

Strengths: Theoretical grounding: the derivation of a link between computation in spiking networks and quadratic convex optimization problems with linear constraints

Weaknesses: - A major weakness is that the paper does not concern with learning of the recurrent weights. Instead, the update rules are for the forward weights and the spiking thresholds of individual neurons. Our understanding of recurrent spiking networks is incomplete, in large part due to the recurrent connections. On the other hand, our understanding of feedforward connections is much more mature. - The results are demonstrated only on toy examples. Recent papers on spiking networks have demonstrated performance values close to those of rate networks, on more complicated tasks. - A central reason for using recurrent architectures is their ability to work with sequential data. In that sense, the provided demonstrations miss a crucial aspect of their computational power.

Correctness: Yes.

Clarity: Yes.

Relation to Prior Work: Yes.

Reproducibility: Yes

Additional Feedback: ------- UPDATE FOLLOWING AUTHOR REBUTTAL LETTER ------- The part of the rebuttal on sequential data is vague and not convincing. Neither of the stated reasons are specific to recurrent connectivity - for static inputs, feedforward networks definitely satisfy those reasons. I liked the delay demonstration in the rebuttal. This demonstration of robustness improves the quality of the paper. (Ideally, the delay should be random and independent for each synapse. e.g., a discrete random process per synapse taking values in {1 ms, 2 ms, 3 ms} at each spike.) Therefore, I decided to increase my score to 5.

[Author Response · NeurIPS 2020]

We first thank the reviewers for their insightful comments which we have taken into careful consideration. We address
the four main areas of criticism below (reviewers referred to as R1-5).

**1. Learning-rule and benchmarking (R1, R2, R5):** *A common critique was that a comparison with state-of-the-art*
*(SOTA) SNNs was lacking. It was also pointed out that learning rules for the recurrent and readout weights were*
*missing.*
If our work were to be evaluated using only performance metrics, this criticism would be fair. However, in machine
learning and science, an equally important goal (and metric) is how well a contribution elucidates underlying principles
of the system under study. We stress that our main objective was to understand the exact nature of computations
done by SNNs and how well this matches biology, rather than improved network performance. We think that we
successfully attained this objective—a large class of SNNs can now be understood as doing non-linear convex input-
output computations within a layer. Moreover, the resulting networks display many biological features such as robustness
and E/I-balance. Given these results, our learning section is simply supposed to illustrate, as a proof-of-concept, that
the problem of learning can be posed differently, once we understand geometrically what the actual computation is.
Learning paradigms for networks of 'convex layers' have been shown to be effective (e.g. Amos & Kolter, 2017), but
are highly non-local and therefore unusable for online learning in SNNs.

**2. Spike-coding network comparison (R2):** *R2 was unsure of the novelty compared to spike coding networks (SCNs),*
*and whether the reported brittleness of SCNs in presence of delays transfers.*
The key advance over standard SCNs is that we show how to perform non-linear computations in these systems.
Standard SCNs such as in Boerlin et al (2013) are restricted to linear computations. (Non-linearities in these networks
were previously introduced by assuming non-linear computations in subthreshold voltages, see e.g., Thalmeier et al,
2016, or Alemi et al, 2018; however, this leads to implausible assumptions on dendritic trees.) The reviewer's second
point, the brittleness of standard SCNs to delays, is indeed well taken and a key criticism of SCNs. Interestingly, our
networks do not suffer from this problem, and are instead very robust to delays, as we explain in Fig. 1. We can update
the paper figures to include this point.

**3. Low-dimensional toy problems (R1, R2, R5):** *Several reviewers had doubts about our focus on single-layered*
*networks solving low-dimensional problems.*
We stress that our main goal was to gain an understanding of the computations performed by a single SNN-layer
dominated by lateral, inhibitory connectivity. It may seem surprising, but such layers are actually not well understood!
We show that the computations done in these layers are far more complex than those of standard LN layers as used in
deep networks. The use of low-dimensional examples was mainly to better communicate the intuitions we gained from
the geometrical picture, and these intuitions do transfer to higher dimensions: SNNs can then be understood as having
a read-out which constantly tries to reach some set-point in the high-dimensional space, while each neuron acts as a
hyper-plane off which the read-out will bounce back into the feasible set.

**4. Recurrent connectivity (R5):** *R5 noted that our networks are missing a crucial aspect of recurrent network*
*computation: working with sequential data.*
Dealing with sequential data is certainly an important application for recurrent connectivity. However, that does not
rule out that recurrent connections may have other functions. In our example, recurrent connectivity is useful because it
(1) helps networks achieve specific input-output functions and (2) allows networks to be robust to many perturbations
(e.g. cell death). We additionally provided a novel way to understand how these two properties and the connectivity are
linked.

Figure 1: (A) (i) In spike-coding networks the connectivity is set up such that it generates a closed feasible space. Delays can then cause several neurons (red + blue) to spike simultaneously, leading to multiple threshold crossings and causing several follow-up spikes (green + gray neurons), and so on, leading to instability. (ii) In our networks, connectivity is set up to lead to an open feasible space. When multiple neurons fire simultaneously due to delays, the system remains within the feasible set, and hence remains stable. (B) The network from Figure 4 in the paper simulated with a delay of 2msec. Note that neither spiking nor functionality are affected by the delay.

[Meta-Review · NeurIPS 2020]

The reviewers expressed some mixed opinions about this work: overall, the idea of interpreting LIF networks as solving quadratic programs (i.e. quadratic objectives with linear constraints) is quite intriguing, but some aspects of the story could be improved. For example, as R5 noted, the synaptic learning rules currently focus on the feedforward weights rather than the recurrent weights. Moreover, I would add that the recurrent weights are subject to relatively strong low-rank assumptions (specifically, GD is rank M, the dimensionality of the variables being optimized, rather than N, the number of neurons/constraints). This property further implies that the diagonal of the recurrent weights, which determine the reset voltage, are also highly constrained. I think this assumption and its implications warrant further discussion. Finally, you say many times that G_i \propto D_i when really you should say G_i \propto (D^T)_i, since on is M \times N and the other is N \times M. Overall, I found the connection between spiking networks and quadratic programming quite interesting, and encourage the authors to address these reviewers' concerns as much as possible for the camera ready.